# Magnetic excitations beyond the single- and double-magnons

Hebatalla Elnaggar [1,2] ✉, Abhishek Nag[3], Maurits W. Haverkort [4], Mirian Garcia-Fernandez[3], Andrew Walters [3], Ru-Pan Wang[1,5], Ke-Jin Zhou [3] ✉ & Frank de Groot[1] ✉

A photon carrying one unit of angular momentum can change the spin angular momentum of a magnetic system with one unit ($\Delta M_s = \pm 1$) at most. This implies that a two-photon scattering process can manipulate the spin angular momentum of the magnetic system with a maximum of two units. Herein we describe a triple-magnon excitation in $\alpha$-$Fe_2O_3$, which contradicts this conventional wisdom that only 1- and 2-magnon excitations are possible in a resonant inelastic X-ray scattering experiment. We observe an excitation at exactly three times the magnon energy, along with additional excitations at four and five times the magnon energy, suggesting quadruple and quintuple-magnons as well. Guided by theoretical calculations, we reveal how a two-photon scattering process can create exotic higher-rank magnons and the relevance of these quasiparticles for magnon-based applications.

Understanding how to control the spin degree of freedom is a cornerstone for several hot topics of contemporary magnetism research, including ultrafast magnetism and magnonics. The main idea behind magnonics is to use elementary magnetic excitations (magnons) for information transfer and processing. Magnons are bosonic quasiparticles and are the quanta of magnetic oscillations of systems with periodically ordered magnetic moments[1]. A magnon is classically depicted as a phase-coherent precession of microscopic vectors of magnetization in a magnetic medium. When a magnon propagates through a magnetic medium, no electrical charge transport is involved, and hence no electrical losses take place. This is the key advantage of using magnons as information carriers. The energy of magnons ranges typically in the terahertz range (in the order of 1–25 THz, i.e., 5–100 meV). The magnon frequency has an important impact on the performance of magnon-based devices because the larger the excitation frequency, the faster the magnons are, at least for a fraction of the magnon band. This means that the use of high-frequency (terahertz) magnons could provide a great opportunity for the design of ultrafast devices[2]. Antiferromagnets represent an appealing playground for the search for new channels of high-frequency, long-lived magnons paving the way toward ultrafast magnon-based devices[3,4].

Collective excitations such as magnons can be effectively measured using $2p3d$ resonant inelastic X-ray scattering (RIXS)[5]. Here a $2p$ core-electron is resonantly excited from its initial state to the empty $3d$ orbitals through an electric dipole transition. This excited state contains a localized core-hole that has a lifetime of ~100 fs determined by Auger decay. The radiative decay of a $2p$ core-hole brings the system back to the ground state, as well as final states, including low-energy excitations. In 1998, it was proposed that $2p3d$ RIXS could be used to measure magnetic excitations, referred to as spin-flip by de Groot and coworkers[5]. Consider a magnetic $3d^9$ system where the hole resides in the $dz^2$ orbital and is spin-up. An incoming photon with the correct energy and polarization can excite a $2p$ core-electron into the empty $dz^2$ spin-up hole leading to a $2p^5 3d^{10}$ intermediate state. While the spin–orbit interaction at the $3d$ shell is in the order of 100 meV, it is ~12 eV for the $2p$ shell: This implies that spin and orbital momenta are mixed in the intermediate state, so neither is

---

[1]Debye Institute for Nanomaterials Science, Utrecht University, 3584 CA Utrecht, The Netherlands. [2]Institute of Mineralogy, Physics of Materials and Cosmochemistry, CNRS, Sorbonne University, 4 Place Jussieu, 75005 Paris, France. [3]Diamond Light Source, Harwell Campus, Didcot OX11 0DE, United Kingdom. [4]Heidelberg University, Philosophenweg 19, 69120 Heidelberg, Germany. [5]Department of Physics, University of Hamburg, Luruper Chaussee 149, G610, 22761 Hamburg, Germany. ✉e-mail: hebatalla.elnaggar@sorbonne-universite.fr; kejin.zhou@diamond.ac.uk; f.m.f.degroot@uu.nl

a good quantum number, and only the total angular momentum is defined. Therefore, the intermediate state can decay to a $3d^9$ with a $dz^2$ spin-down hole final state. This excitation is referred to as a spin-flip excitation because when one compares the initial state with the final state, one finds that the only change between both is the spin projection from up to down and a corresponding counter change in the photon angular momentum. In a magnetic material such as $\alpha$-$Fe_2O_3$, the magnetic excitations are nonlocal collective excitations in the form of magnons. A magnon excitation is commonly interpreted to originate from a local single-site spin-flip RIXS process. This description is widely used because the intermediate state in a $2p3d$ RIXS experiment is strongly localized due to the $2p$ core-hole, and hence such a local picture is very useful to describe many aspects of the RIXS process. However, the final state excited by RIXS is not necessarily localized and can be a collective excitation such as magnons in a magnetic material. The way to reconcile both the local single-site and collective aspects of RIXS is to realize that the incident photon can be scattered at any equivalent site, leading to a final state that is a superposition of spin-flips at equivalent sites. Such a final state carries a nonlocal magnetic excitation and represents the magnon density of states, as also confirmed by detailed comparison to inelastic neutron scattering (INS) data[6]. As a photon in the X-ray regime has a non-negligible linear momentum, one can also measure the dispersion of magnons when spin-flip scattering is allowed. This realization has been the main motivation behind the development of high-resolution RIXS beamlines, with the goal of studying the spin dynamics of (pseudo)spin S = ½ materials such as cuprates and iridates[7–10].

Spin-half systems represent a special case as only transitions from $M_s = -\frac{1}{2}$ to $M_s = \frac{1}{2}$ are allowed on a single atomic site. These excitations propagate a change of one unit of angular momentum and are similar to the magnons observed by other techniques such as INS and Raman scattering[11]. We will refer to these magnons as conventional single-magnons. Whereas it is only possible to flip a single spin at a local single magnetic site for cuprates and iridates, resulting in a collective single-magnon excitation in the material, the nickelates can theoretically present single and double spin-flip excitations. This is because the $Ni^{2+}$ is $3d^8$ with S = 1, and hence excitations between $M_s = 1$, 0, −1 are possible on a single site leading to single- and double-magnons in the extended system. RIXS measurements on NiO have confirmed the presence of single ($\Delta M_s = 1$) and double-magnons ($\Delta M_s = 2$) in the system[12,13]. We point out that double-magnons are different from bimagnons observed in cuprates. A double-magnon is a $\Delta M_s = 2$ transition, while a bimagnon is composed of two single-magnons, one changing the spin projection with +1 (i.e., $\Delta M_s = 1$) and the other with −1 (i.e., $\Delta M_s = -1$) giving rise to a combined $\Delta M_s = 0$ transition[14].

While it is clear that for high spin $Ni^{2+}$ ions in NiO possessing two unpaired $3d$ electrons, only two spins can change their angular momenta (i.e., excitations between $M_s = 1$, 0, −1), the situation is more complicated for a high spin $3d^5$ ion in a magnetic system. In this case, there are conceptually five spins that could be locally reversed, resulting in magnons carrying a change of angular momentum of up to 5 units in the extended system (i.e., local spin-flip excitations between $M_s = 5/2$, 3/2, ½, −½, −3/2, −5/2 leading to a higher-rank magnon final state that is a superposition of higher-rank spin-flips at equivalent sites). This raises the fundamental question: Is it possible to change the spin angular momentum of a system with an amount greater than the change in the X-ray photon angular momentum of the RIXS experiment?

Here we provide experimental data capable of answering this question by measuring the low-energy magnon spectrum of an $\alpha$-$Fe_2O_3$ single crystal at the ultrahigh-resolution I21 RIXS setup ($\Delta E = 32$ meV) at Diamond Light Source[15]. Guided by theory, we show that the crystal lattice acts as a reservoir of angular momentum which provides the extra angular momentum required to excite the higher-rank magnons (i.e., beyond single- and double-magnons). We developed a low-energy effective operator that describes the higher-rank magnons and derived simple selection rules that can be used to predict the best experimental settings for exciting the higher-rank magnons.

## Results

The antiferromagnet $\alpha$-$Fe_2O_3$ is an ideal material to initiate this kind of study because the ground state of $Fe^{3+}$ has the maximum number of unpaired electrons for the $d$ orbitals providing a platform to test the maximum number of possible spin-flip excitations. Furthermore, the $^6A_1$ orbital singlet ground state makes a clean case to study solely spin excitations without any orbital contribution.

Figure 1a shows the Fe ions in the unit cell of $\alpha$-$Fe_2O_3$. The exchange coupling is dominantly antiferromagnetic with the Néel temperature $T_N$ of ~950 K. In addition to the Néel transition, $\alpha$-$Fe_2O_3$ exhibits another magnetic transition, referred to as the Morin transition ($T_M$ of ~250 K), where below $T_M$, it is purely antiferromagnetic. We performed our measurements at 13 K ($T < T_M$) in the collinear antiferromagnetic phase. An exemplary $L_3$ X-ray absorption spectrum (XAS) is shown in Fig. 1b, where we find two main peaks (labeled $E_1$ and $E_2$, where this splitting is due to the crystal field) exhibiting the expected X-ray magnetic linear dichroism signal in line with the previous literature[16–18].

The RIXS spectrum measured at $E_1$ is shown in Fig. 1c. The elastic line is observed at zero-energy transfer, where the final state preserves the initial state spin orientation. A cascade of energy transfer peaks can be seen at 100, 200, 300, 400, and 500 meV. The first energy transfer peak can be assigned to a single-magnon excitation. This agrees well with the observation from INS experiments, where an optical nearly non-dispersing mode is observed at ~100 meV[19]. The single-magnon excitation propagates a change of angular momentum of $\Delta M_s = 1$. The second energy transfer peak appears at double the energy of the single-magnon and can be assigned to a double-magnon excitation ($\Delta M_s = 2$) similar to the double-magnon excitation observed in NiO[12,13]. The most remarkable feature in our results is the ability of the two-photon RIXS process to excite higher-rank magnons at 3 times, 4 times, and potentially at 5 times the energy of a single-magnon (zoom in Fig. 1d and its first derivative in Fig. 1e). These higher-rank magnons propagate these multiples of angular momentum. We provide the details of the fitting and the full energy loss spectra at two incident energies in the materials and methods in Supplementary Figs. S1, S2, and S3.

We measured the angular dependence of the single-, double- and triple-magnons by rotating the sample in the azimuthal ($\alpha$) direction to decipher the nature of the transitions involved (see Fig. 2). Conceptually, one expects that the angular behavior of the higher-order magnons differs from the single-magnon as the angular momentum selection rules are different ($\Delta M_s = 1$, 2, 3 involves a dipolar, quadrupolar, and hexapolar spin-flip process respectively). This is confirmed by comparing the angular behavior in Fig. 2a–c, where the magnitudes of the transitions are reduced approximately with an order of magnitude as we move from single- to double- to triple-magnons, in addition to the change of the angular profile. The angular-dependent RIXS of these excitations, however, did not show any noticeable dispersion (see Supplementary Fig. S4). The parent single-magnon is an optical mode that shows negligible dispersion according to INS measurements which agrees with our measurement[19].

## Discussion

The implication of our experimental observation is that the two-photon RIXS process can exchange five units of angular momenta with the magnetic material. This is an unexpected result because

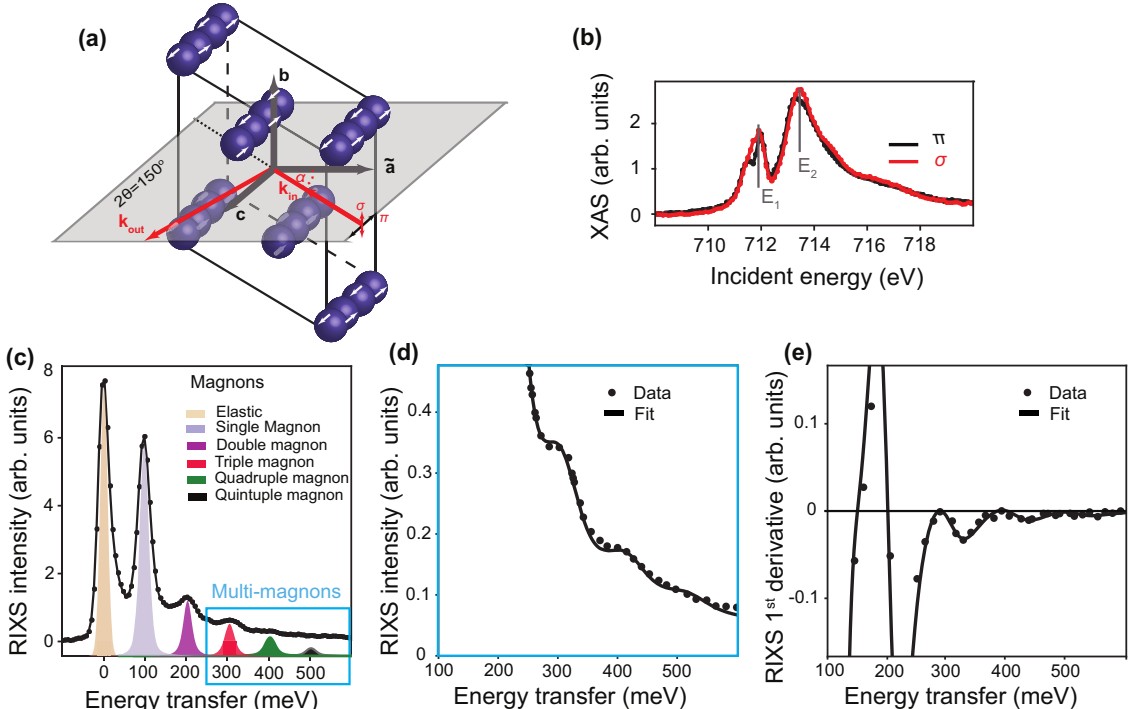

**Fig. 1 | Crystal structure and X-ray $L_3$-edge measurements in a (0001) $\alpha$-$Fe_2O_3$ single crystal. a** Crystal structure of $\alpha$-$Fe_2O_3$ showing only the Fe atoms and the scattering geometry used for all measurements presented in this work. $k_{in(out)}$ are the incident and outgoing wave vectors, and the scattering angle (2θ) was kept fixed at 150°. The incidence angle is $\alpha$ with $\alpha = 90°$ for normal incidence. The antiferromagnetic order is depicted with white arrows showing the orientation of the magnetic moments. **b** Fe $L_3$ XAS measured with $\pi$ (black) and $\sigma$ (red) polarization. Two main peaks can be identified and are labeled $E_1$ and $E_2$. **c** RIXS spectra measured at $E_1$ ($\alpha = 95°$, $\pi$ polarization). The orange-shaded Gaussian peak at zero-energy transfer corresponds to the elastic peak, also having a contribution from $\Delta M_S = 0$ excitation. The five shaded antisymmetric Lorentzian peaks represent the single- (blue), double- (purple), triple- (red), quadrupole- (green), and quintuple- (black) magnon excitations (see "Methods" for the fitting details). The blue box highlights the spin non-conserving transitions. **d** A zoom on the spin non-conserving transitions. The dots are experimental data, and the black line is the fit. The triple- and quadrupole-magnons can be clearly identified. **e** The first derivative of (**d**) where the signal equals zero at the position of the higher-rank magnons.

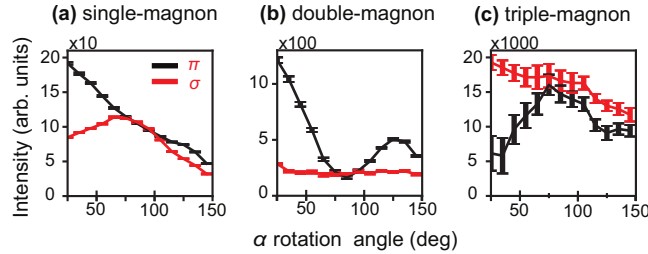

**Fig. 2 | Angular dependence of the magnons measured with $\pi$ and $\sigma$ polarization. a** Single-magnon, **b** double-magnon, and **c** triple-magnon excitations measured at the incidence energy $E_1$. The angular dependence is measured by rotating the single crystal in the azimuthal direction ($\alpha$ rotation) while the scattering angle was kept fixed at 150°. The error bars shown are the least square fitted intensity value errors.

the selection rule for a dipole transition in the presence of strong spin-orbital coupling is that the change of the total angular momentum ($\Delta M_j$) is equal to 0 or ±1. This means that for a dipole-in ($2p\rightarrow 3d$ transition), dipole-out ($3d\rightarrow 2p$ transition) $2p3d$ RIXS process, the possible transitions should involve $\Delta M_j = 0, \pm1, \pm2$ giving rise to only single- and double-magnons. We performed multiplet ligand-field theory calculations for $Fe^{3+}$ $L_3$-edge RIXS (see Fig. 3a), which confirms that only single- and double-magnons are expected to be observed and is in line with previous work on NiO[12,13]. It is essential to examine the interaction terms of the model Hamiltonian responsible for the single- and double-magnons to find the origin behind the higher-rank

magnons in $\alpha$-$Fe_2O_3$.

$$H = \sum_k f_k F^k + \sum_k g_k G^k + \sum_i l_i \cdot s_i + J_{exch}(n.S) \quad (1)$$

The model Hamiltonian used for the calculation is given by Eq. 1. The $J_{exch}(n.S)$ term is the mean-field super exchange interaction term that determines the energy of the single-magnon. The spin–orbit coupling is given by the $\sum_i l_i \cdot s_i$ term and is responsible for the spin-flip process by mixing the orbital and spin degrees of freedom and enables the observation of single-magnons as detailed in the work of de Groot et al.[5]. The double-magnon is enabled through the intra-atomic Coulomb exchange given by $\sum_k f_k F^k + \sum_k g_k G^k$. $F_k(f_k)$ and $G_k(g_k)$ are the Slater–Condon parameters for the radial (angular operators) part of the direct and exchange Coulomb interactions, respectively. The intra-atomic Coulomb exchange interaction strongly couples the valence and core electrons, implying that the spin angular momentum of both the core and valence orbitals are no longer a good quantum number, effectively leading to $\Delta M_S = 0, 1$, and 2 excitations.

It is inevitable to conclude that the higher-rank magnons have a different origin compared to the single- and double-magnons reported in previous works[12,13,20], and a mechanism that allows the exchange of higher angular momenta is needed. This apparent contradiction can be reconciled by realizing that the angular momenta of electrons only is not a conserved quantum number in real crystals[21]. The crystal lattice can exchange angular momentum with the electrons providing the extra angular momentum required for higher-rank magnons involving $\Delta M_S > 2$. We performed a full-

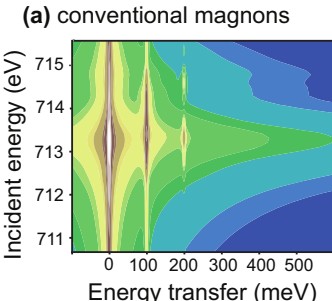

**(a) conventional magnons**

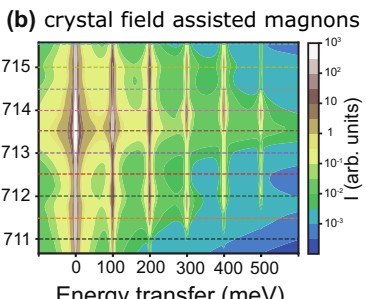

**(b) crystal field assisted magnons**

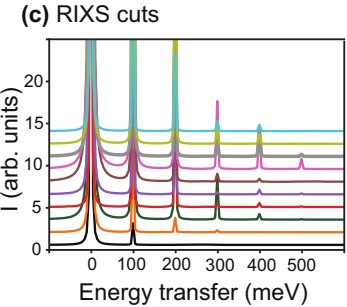

**(c) RIXS cuts**

**Fig. 3 | Full-multiplet ligand-field theory calculation for $Fe^{3+}$ $L_3$-edge RIXS in α-$Fe_2O_3$. a** Incident energy-dependent RIXS intensity map for a $Fe^{3+}$ ion according to the Hamiltonian in Eq. 1. **b** Considering an additional term that includes the crystal field effects. The calculations were performed for linear horizontal incoming beam and unpolarized outgoing beam to correspond to the experimental conditions. The parameters used for the calculations are reported in Supplementary Table 1. We note that the intensity is plotted in a logarithmic scale. **c** Line cuts through the RIXS map of (**b**) plotted at the dashed lines positions shown on the map.

multiplet ligand-field theory calculation for $Fe^{3+}$ $L_3$-edge RIXS, including the effect of the crystal lattice using an effective octahedral crystal field potential (see Fig. 3b). In addition to the single- and double-magnons, triple-, quadruple-, and quintuple-magnons are now visible confirming that the crystal field potential is the key factor behind the higher-rank excitations as can be seen in details in the linear cuts of the RIXS map in Fig. 3c. Here we stress that the higher-rank magnons are generated from a single magnetic site, and the role of the crystal lattice can be considered as a reservoir of angular momentum supplying the extra angular momentum required.

To visualize the role of the crystal lattice, we follow in Fig. 4 the fate of an excitation created by the absorption, for example, of a circular right-polarized photon. We define the polarization of light as: $|R\rangle = \frac{1}{\sqrt{2}}[1,-i,0]$, $|L\rangle = [1,i,0]$ and $|Z\rangle = [0,0,1]$. The initial state can be represented by the vector shown in Fig. 4a, which includes the 2p and 3d orbitals participating in the RIXS process. The first two numbers shaded in gray represent the occupation of the 2p spin-down (red arrow) and 2p spin-up orbitals (blue arrow), where we take the spin quantization axis to be the $C_4$ axis of the octahedron. The second two numbers shaded in yellow are the 3d spin-down (red arrow) and 3d spin-up orbitals (blue arrow) occupation numbers. The projection of the total orbital angular momentum, $L_Z$, is specified in the subscript of the vector to keep track of the orbital angular momentum of the states. This means that the initial state is given by $(3,3,5,0)_0$.

Upon the absorption of the polarized photon, a spin-up electron can be excited from the $2p_{-1}$ to the $3d_{-2}$ orbital resulting in an intermediate state given by $(3,2,5,1)_{-1}$ (Fig. 4a). This intermediate state can decay back elastically by emitting a right-polarized photon contributing to the elastic RIXS peak (Fig. 4b1). Another possible path is shown in Fig. 4b2, where a $3d_{-2}^{\uparrow}$ electron scatters off the crystal field potential to a $3d_2^{\uparrow}$ orbital and thereby changes its angular momentum by four units. This extra angular momentum provided by the lattice is the key aspect that makes it possible to excite higher-rank magnons. A cascade of 2p spin−orbit coupling and 2p-3d exchange interaction is required to transfer this orbital angular momentum to spin angular momentum, as illustrated in Fig. 4c−h. The first pair of 2p spin−orbit coupling and 2p-3d exchange interaction changes the intermediate state to $(3,2,4,2)_2$. We note that this intermediate state cannot decay to a single-magnon excitation as it cannot reach an $L_Z = 0$ final state through a dipole emission. The second pair of 2p spin−orbit coupling and 2p-3d exchange interaction changes the intermediate state to $(3,2,3,3)_1$. Now this intermediate state can decay to a double-magnon excitation either after the 2p spin−orbit coupling step ($2p_{-1}^{\downarrow} \to 3d_{-2}^{\downarrow}$ emitting a left-polarized photon−not shown in Fig. 4 for visual clarity) or after the exchange interaction step emitting a left-polarized photon (Fig. 4g1). A final 2p spin−orbit coupling step is required to

change the intermediate state to $(2,3,3,3)_0$, which can finally decay to a triple-magnon by emitting a Z-polarized photon (Fig. 4h).

Quadruple and quintuple-magnons can be reached by a further exchange of angular momentum with the lattice followed by cascades of 2p spin−orbit coupling and 2p-3d exchange interaction. In contrast, when the crystal field is not considered, only single- and double-magnons can be excited (see the Feynman diagrams in Supplementary Fig. S5). The transparent Feynman diagram representation allows us to derive selection rules for this example: (1) it is not possible to excite single-magnons using circular right-polarized incoming X-rays, (2) double-magnons can be selectively observed by detecting the left-polarized outgoing light. (3) triple-magnons can be selectively observed by detecting the Z-polarized outgoing light. The Z-polarized light can be experimentally detected by placing an extra detector in the vertical plane, for example. A full RIXS calculation is shown in Supplementary Fig. S6 and confirms the selection rules derived from Fig. 4.

We developed a low-energy effective RIXS operator that describes low-energy magnetic excitations such as magnons in terms of spin operators based on the work of Haverkort[22]. The full RIXS cross-section is given by Eq. 2, where the ground state $|i\rangle$ is excited by a photon described by a dipole transition operator $T_{\epsilon_i}$ to an intermediate state described by the Hamiltonian (H) and decays to all possible final states $|f\rangle$ emitting a photon described by a dipole transition operator $T_{\epsilon_o}$.

$$RIXS(\omega) \propto \sum_f \left| \left\langle f|T_{\epsilon_o}^{\dagger} \frac{1}{\omega_i - H + i\Gamma/2} T_{\epsilon_i}|i\right\rangle \right|^2 = \sum_f |\langle f|R_{eff}|i\rangle|^2 \quad (2)$$

The effective operator ($R_{eff}$) removes the intermediate state from the equation by expanding the intermediate state Hamiltonian in terms of polynomials of spin operators multiplied by X-ray absorption fundamental spectra. The expression of the expansion to the third order is presented in the materials and methods and is summarized in Fig. 5a. The spin-flip processes resulting in magnons can be grouped in order of the spin operator rank: linear (shaded in blue), quadratic (shaded in purple), and cubic (shaded in red) operators. The linear spin operators can generate single-magnons while the quadratic spin operators generate single- and double-magnons and produce the majority of the RIXS intensity (see Fig. 5b). Finally, the cubic spin operators can generate single-, double- and triple-magnons. The main advantage of this expansion is its simple form that allows one to determine general selection rules depending on the polarization of the incoming and outgoing light.

We computed the angular dependence of the single-, double- and triple-magnons based on this expansion in Fig. 6. Our calculations capture the essential aspects of the experimental

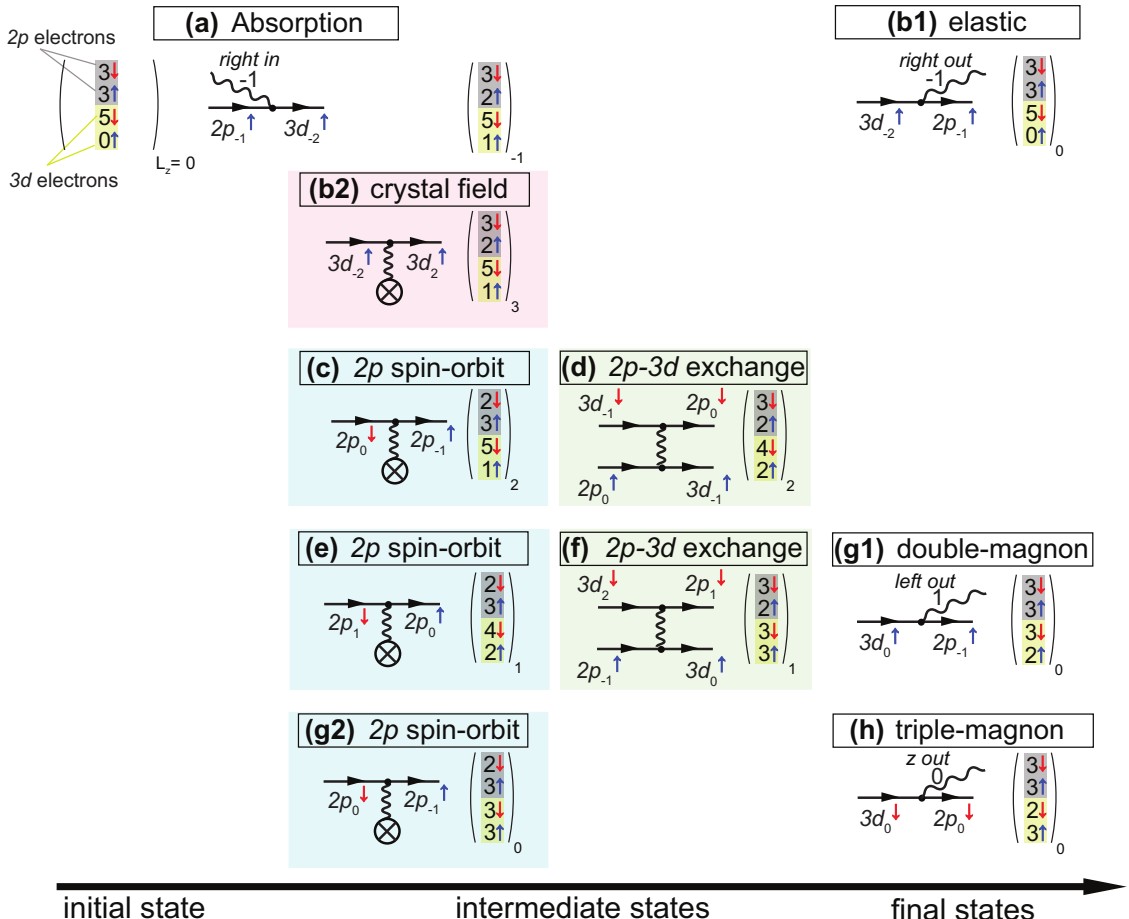

**Fig. 4 | Schematic representation of the mechanism of higher-rank magnons by 2p3d RIXS.** The initial state vector is shown in the upper left corner comprising the 2p (gray) and 3d (yellow) orbitals participating in the RIXS process. The spin of the electrons is depicted by the colored arrows (red = down, blue = up). We follow the fate of a 2p→3d excitation created by the absorption of a right-polarized photon (**a**) up to the triple-magnon decay through a cascade of crystal field interaction, 2p spin–orbit coupling, and 2p-3d exchange interaction through the steps from (**b–h**).

angular dependence where we obtain the correct order of magnitude for the magnons and reproduce the general angular dependence confirming the nature of the higher-rank magnons (compare Fig. 2 to Fig. 6). Some deviations of the calculated angular dependence from the experiment could be to several factors. On the experimental side, our measurements were performed on a bulk single crystal which is prone to self-absorption and saturation effects. As the RIXS cross-section is a combination of an absorption (photon-in) process and an emission (photon-out) process, two geometrical effects have to be taken into account here: the probing depth is dependent on the X-ray absorption spectroscopy (XAS) cross-section (saturation), and the emitted photons can be re-absorbed (self-absorption). Consequently, the RIXS intensity is distorted in a bulk crystal according to the photon energy and the experimental geometry[23]. When the sample is rotated, the probing depth is changed, and the photons emitted at different energies have different escape lengths, hence distorting the angular dependence. It is difficult to correct for these geometrical energy-dependent effects because it is affected by the background absorption (which is the off-resonant contribution from other elements in the sample and in the path of the beam). On the theoretical side, one likely reason for the deviation could be the fact that the Fe sites in hematite have a small trigonal distortion. As a first approximation, the trigonal distortion would not change the ground state of $Fe^{3+}$ because the singlet $^6A_1$ ground state does not split. However, the trigonal distortion

would influence the intermediate states and hence could modify the intensity and, consequently, the angular dependence[18]. Finally, we point out that ligand-field multiplet theory reduces a full solid to a local cluster. This means that any Fe–Fe interactions or intercluster hopping are not considered. We expect that the above approximations can affect the angular dependence[24].

The comparison between theory and experiment on α-$Fe_2O_3$ confirms that the cascade of excitations we observed at triple, quadruple, and quintuple the energy of a single-magnon are higher-rank-magnons that propagate these higher multiples of spin angular momentum. From a fundamental point of view, the higher-rank magnons can couple differently with the various degrees of freedom of the system, providing a unique platform to investigate magnon interactions. From a technological point of view, these excitations have higher energies than that of single-magnons and hence are potentially more thermally robust. We predict that these higher-rank magnons can also be excited using THz-pulses and can be enhanced using magnonic crystals, paving the way toward future magnonic devices.

## Methods

### Resonant inelastic X-ray scattering measurements

High-resolution (ΔE = 32 meV), Fe $L_3$-edge resonant inelastic X-ray scattering measurements were done at the I21 beamline of the Diamond Light Source, United Kingdom. Linear horizontally (π) or vertically polarized (σ) X-ray beam was used. The angular

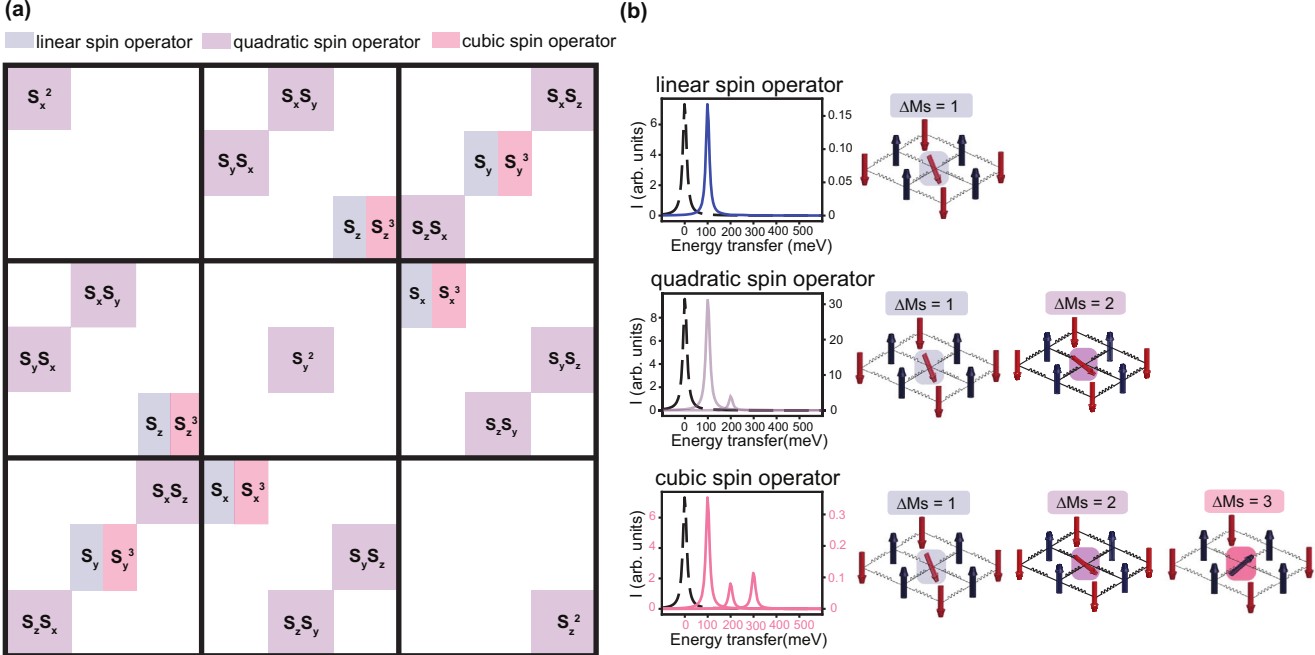

**Fig. 5 | Matrix elements of the effective RIXS operator ($R_{eff}$) expanded up to the third rank and the RIXS spectrum from individual operators. a** Summary of the expansion presented in Eq. 3 grouped in terms of linear (blue), quadratic (purple), and cubic (red) spin operators (**S**) involved in the RIXS cross-section. **b** The RIXS cross-section computed using the three orders of the spin operators.

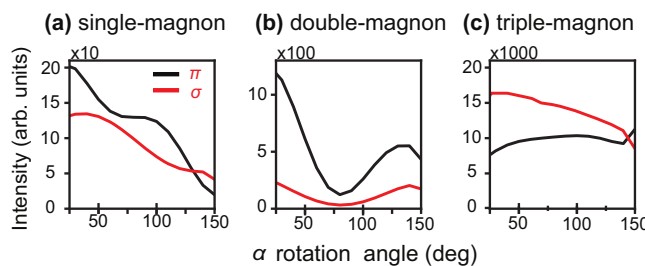

**Fig. 6 | Computed angular dependence of the magnons measured with $\pi$ and $\sigma$ polarization using the effective operator expansion. a** Single-magnon, **b** double-magnon, and **c** triple-magnon excitations at the incidence energy $E_1$.

dependence was measured by rotating the sample about the $b$-axis (referred to as α rotation) of a polished α-$Fe_2O_3$ single crystal cooled down to 13 K. The Morin temperature of $Fe_2O_3$ is 220 K. The scattering angle was kept fixed at 150°. The X-ray absorption spectrum (XAS) shown in Fig. 1b was measured using total electron yield in the same geometry. The pressure in the experimental chamber was maintained below $5 \times 10^{-10}$ mbar. The zero-energy transfer position and resolution of the RIXS spectra were determined from subsequent measurements of elastic peaks from an adjacent carbon tape.

### Resonant inelastic X-ray scattering data fitting
RIXS data were corrected for self-absorption prior to fitting. The elastic line was fitted with an energy resolution limited Gaussian lineshape (orange shade, Fig. 1c and Supplementary Fig. 1a). The phonon peaks close to 43 meV and 150 meV were fitted with asymmetric Lorentzian functions (shown by gray dashed lines) convoluted with the energy resolution. These peaks are clearly visible in RIXS spectra at the $E_2$ peak of XAS (see inset of Supplementary Fig. 1b). The five shaded antisymmetric Lorentzian peaks convoluted with the energy resolution represent the single- (blue), double- (purple), triple- (red), quadruple-

(green), and quintuple- (black) magnon excitations (Fig. 1c and Supplementary Fig. 1a). While the energy positions up to the quadruple magnon excitation were kept as a free parameter for fitting, the energy position of the quintuple magnon was fixed to (fitted energy position of the triple-magnon)*(5/3). See also Supplementary Figs. 2 and 3 for the low energy fits to the RIXS data for different α at $\pi$ and $\sigma$ polarizations, respectively.

### Multiplet ligand-field calculations
The crystal field multiplet model is an effective model Hamiltonian for the description of all charge-conserving excitations of ionic transition metal systems. The crystal field multiplet model is valid for the main peaks of $2p$ X-ray absorption and the low-energy RIXS excitations of ionic transition metal ions, because the $2p3d$ X-ray absorption and the $3d2p$ X-ray emission are neutral, self-screened, transitions, which implies that screening channels such as ligand-metal charge transfer can be approximated by renormalized parameters. We used the quantum many-body program Quanty[25] to simulate Fe XAS and $2p3d$ RIXS in α-$Fe_2O_3$. The Hamiltonian used for the calculations consists of the following terms: (1) Coulomb interaction, (2) crystal field potential, (3) spin–orbit coupling, and (4) effective exchange interaction. The $d$–$d$ ($p$–$d$) multipole part of the Coulomb interaction was scaled to 70% (80%) of the Hartree–Fock values of the Slater integral. The general parameters used for the simulations agree with previous studies of α-$Fe_2O_3$ $L_{2,3}$ edges.

### Expression of the effective RIXS operator
The effective operator can be expressed by Eq. 3, as shown by Haverkort[22]. Here $\epsilon_{in(out)}$ is the incoming (outgoing) polarization of the photons. $F_{\{x,y,z\}}$ is the conductivity tensor describing the full magneto-optical response function of the system depending on the local magnetization direction given by $\{x,y,z\}$.

$$R_{eff} = -\text{Im}[\epsilon_{in}^* \cdot F_{\{x,y,z\}} \cdot \epsilon_{out}] \quad (3)$$

The general form of the conductivity tensor can be expressed as a sum of linear independent spectra multiplied by functions depending on the local magnetization direction, as given in Eq. 4.

$$F_{\{x,y,z\}} = \sum_{k=0}^{k} \sum_{m=-k}^{k} \begin{pmatrix} F_{xx}^{k,m} & F_{xy}^{k,m} & F_{xz}^{k,m} \\ F_{yx}^{k,m} & F_{yy}^{k,m} & F_{yz}^{k,m} \\ F_{zx}^{k,m} & F_{zy}^{k,m} & F_{zz}^{k,m} \end{pmatrix} Y_{k,m}(\theta,\phi) \qquad (4)$$

Here $\theta$ and $\phi$ define the direction of the local moment with $\theta$ being the polar angle, and $\phi$ being the azimuthal angle. $Y_{k,m}(\theta,\phi)$ is a spherical harmonic function and $F_{ij}^{k,m}$ is the $i,j$ component of the conductivity tensor on the basis of linear polarized light in the coordinate system of the crystal. In symmetries lower than spherical, this expansion on spherical harmonics does not truncate at finite $k$ and the angular momentum of the electrons only is not a conserved quantum number in the crystals. We have shown in our previous work that including terms up to $k = 3$ is sufficient to describe $Fe^{3+}$ ions in octahedral crystal field[26]. The expression is given in Eq. 5 and involves terms up to the third order in spin leading to single, double, and triple spin-flip processes.

$$F_{\{x,y,z\}} = \begin{pmatrix} F_{a1g}^0 + 2F_{eg}^2(S_x^2 - \frac{1}{3}S^2) & F_{t2g}^2(S_xS_y + S_yS_x) - F_{t1u}^1Sz - F_{t1u}^3(-\frac{3S_z}{5} + S_z^3) & F_{t1u}^1S_y + F_{t1u}^3(-\frac{3S_y}{5} + S_y^3) + F_{t2g}^2(S_xS_z + S_zS_x) \\ F_{t2g}^2(S_y + S_yS_x) + F_{t1u}^1Sz + F_{t1u}^3(-\frac{3S_z}{5} + Sz^3) & F_{a1g}^0 + 2F_{eg}^2(S_y^2 - \frac{1}{3}S^2) & -F_{t1u}^1S_x - F_{t1u}^3(-\frac{3S_x}{5} + S_x^3) + F_{t2g}^2(S_yS_z + S_zS_y) \\ -F_{t1u}^1S_y - F_{t1u}^3(-\frac{3S_y}{5} + S_y^3) + F_{t2g}^2(S_xS_z + S_zS_x) & F_{t1u}^1S_x + F_{t1u}^3(-\frac{3S_x}{5} + S_x^3) + F_{t2g}^2(S_yS_z + S_zS_y) & F_{a1g}^0 + 2F_{eg}^2(S_z^2 - \frac{1}{3}S^2) \end{pmatrix} \qquad (5)$$

## Data availability

The data generated and analyzed are included in the paper and its supplementary information and have all been deposited in the Zenodo database under the accession code at https://zenodo.org/record/7828290. Raw data files will be made available upon request.

## Code availability

The code that supports the findings of this study is available online and can be downloaded at: https://www.quanty.org, together with full documentation and instructions to use it. The script file required to reproduce the theoretical figures has been deposited in the Zenodo database under the accession code at https://zenodo.org/record/7828290.

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

## Acknowledgements

We acknowledge the staff of beamline I21 of Diamond Light Source for their help in setting up and running the experiments. We acknowledge the following fundings: Dutch Research Council Rubicon Fellowship Project No. 019.201EN.010 (H.E.), ERC advanced Grant XRAYonACTIVE No. 340279 (F.d.G.).

## Author contributions

Conceptualization: H.E., F.d.G. Methodology: H.E., F.d.G., R.W., A.N., K.J.Z. Investigation: A.N., K.J.Z., M.G.F., A.W. Visualization: H.E., F.d.G., M.W.H. Writing—original draft: H.E. Writing—review & editing: H.E., F.d.G., M.W.H., K.J.Z.

## Competing interests

The authors declare no competing interests.
