## [Peer Review File · Nature Communications]

Reviewers' Comments:

Reviewer #1:

Remarks to the Author:

Dear Editor,

the authors addressed my previous questions. But I still have difficulties following the authors reasoning on the difference between local, single-site excitations of electrons and nonlocal, collective excitations such as magnons.

Maybe the reason of my confusion is because we are from a different community with different usage of words. Nevertheless, I think that the manuscript should be understandable for a larger community if published in *Nature Communications*.

Hence, I have a question where I am believe that a good answer could create a bridge to a larger readership: The authors raise a fundamental question (Line 83-85): „Is it possible to change the spin angular momentum of a system with an amount greater than the change in the x-ray photon angular momentum of the RIXS experiment?“

In my research field, we also have multi-magnon scattering processes, where the multiple magnons scatter and annihilate and create new magnons. One famous example is three-magnon-scattering, where a the ferromagnetic resonance can decay into two new magnons with half the energy and opposite linear momentum. The spin momentum, however, is obviously not conserved (1 magnon in, two magnons out). It is generally accepted, that the break in rotational symmetry due to the non-isotropic magnetic-dipole-interaction is causing the breakdown of angular momentum conservation. Also, keep in mind that a magnon is not only the excitation of the spin-system. Even magnons with high wave-vectors are still polaritons, meaning they are a hybridized excitation of a non-local, collective spin flip AND an electromagnetic wave coupled to the magnetic excitation.

Here is my fundamental question: If multi-magnons are excited in the authors experiment and if this is indeed a non-local, collective excitation of several electrons, then there should be a magnetic-dipole field associated with this excitation, which also breaks the angular momentum conservation in scattering processes with X-rays.

Reviewer #2:

Remarks to the Author:

The updated version of the manuscript by Elnaggar et al. is now submitted to *Nat. Comm.* after the first round of review. They report on high rank magnon excitation from RIXS experiments.

I acknowledge that the authors do address well concerns on the existence of these 3rd, 4th magnons in the experimental data. This seems to be better valid on reporting the derivative of the data.

However, my concern on the agreement between the experiment and theory still holds. The authors comments on my point (3) in the previous report does not solve the question. They argue that the high rank magnon excitations intensities are maximized at specific incident energy in the experimental data. That would be fine, however looking at simulated data (color RIXS map) for the Incident and energy transfer, the high rank magnons do show up at several incident energies. To me this simulation result is still in disagreement with the experimental data.

Moreover, I find their argument "We removed the cuts measured at E2 from the supplementary as it is confusing and does not add extra information." not adequate in this case. This seems to be hiding experimental data that disagrees with their conclusion. I judge this to be very inadequate

for a scientific publication of this importance.

Their argument on the agreement between the theory and experiment for the angular dependence also does not convince me. It is fine that the theoretical model include many approximations to be close to the experimental data, however these approximations must always be reported. If the final version still does show marginal similarity between the experimental and theoretical data, this is a point that has to be questioned. To me they cannot claim their conclusions with this disagreement between theory and experiment.

Overall, I do see a nice report of the experimental observation of the 3rd and 4th magnons. However I do see disagreement between the theory and the experiments, which make their bold conclusions to be questionable. Also, I find their decision to remove the experimental data that brings up these questions to be very inadequate.

It is my assessment that the manuscript should not be accepted for publication in Nature Communications. Perhaps the authors should bring back all experimental data to the supplementary materials and address that the simulations do have some limitations to reproduce the experimental data. In this case, I would find the manuscript to be suitable to a much more specialized journal.

Reply to Reviewer comments on:

Novel magnetic excitations beyond the single- and double-magnons

Hebatalla Elnaggar,^{1,2*} Abhishek Nag,³ Maurits W. Haverkort,⁴ Mirian Garcia-Fernandez,³
Andrew Walters,³ Ru-Pan Wang,¹ Ke-Jin Zhou,^{3*} Frank de Groot^{1*}

¹Debye Institute for Nanomaterials Science, Utrecht University, 3584 CA Utrecht, The Netherlands.

²Institute of Mineralogy, Physics of Materials and Cosmochemistry, Sorbonne University, 4 Place Jussieu, 75005 Paris, France.

³Diamond Light Source, Harwell Campus, Didcot OX11 0DE, United Kingdom.

⁴Heidelberg University, Philosophenweg 19, 69120 Heidelberg, Germany.

Reviewer comments are in black

Authors reply is in blue

Changes made are in red and are highlighted in the Manuscript_TrackChanges file

REVIEWER COMMENTS

Reviewer #1 (Remarks to the Author):

The updated version of the manuscript by Elnaggar et al. is now submitted to Nat. Comm. after the first round of review. They report on high rank magnon excitation from RIXS experiments. I acknowledge that the authors do address well concerns on the existence of these 3rd, 4th magnons in the experimental data. This seems to be better valid on reporting the derivative of the data.

However, my concern on the agreement between the experiment and theory still holds. The authors comments on my point (3) in the previous report does not solve the question. They argue that the high rank magnon excitations intensities are maximized at specific incident energy in the experimental data. That would be fine, however looking at simulated data (color RIXS map) for the Incident and energy transfer, the high rank magnons do show up at several incident energies. To me this simulation result is still in disagreement with the experimental data.

The scale bar of the RIXS color map is logarithmic and that can convey the impression that the higher-rank magnons show up at many incident energies. Below are the RIXS cuts at a mesh of incident energies. Fig. 1B shows that the multi-magnons have negligible spectral weight at many incident energies but are enhanced at particular energies. We have experimentally measured RIXS at two incident energies (at the maxima of the two peaks of the L₃ edge E1 and E2). The calculations predict that the measurement at E1 enhances the multi-magnons while the measurement at E2 does not enhance the multi-magnons. This agrees with what we observe experimentally. Our experimental data do not imply that we can only observe the multi-magnons at a single energy (E1).

Figure 1: A) Calculated Fe^{3+} RIXS map in Fe_2O_3 . The dashed lines show the energy cuts plotted in (B). The cuts are color coded according to the RIXS map. (C) RIXS cuts at the two incident energies E_1 and E_2 which we measured experimentally.

We noted that the intensity is plotted in logarithmic scale in the caption of Fig. 3 line 201.

Moreover, I find their argument "We removed the cuts measured at E2 from the supplementary as it is confusing and does not add extra information." not adequate in this case. This seems to be hiding experimental data that disagrees with their conclusion. I judge this to be very inadequate for a scientific publication of this importance.

We apologize for this: it was not our intension to hide experimental data. In our judgment, the cut at E2 does not show the multi-magnons because the incident energy that we used in the measurements was chosen at the peak maximum of the x-ray absorption, which turns out to give a minimum for the multi-magnon excitations, as shown both in experiment and in theory.

We have put back the cuts measured at E2 to the supplementary and we have taken this as an opportunity to explain the importance of energy tuning in Fig. S3 of the supplementary information. Fig. S3 is now referenced in the manuscript is lines 141-142.

Their argument on the agreement between the theory and experiment for the angular dependence also does not convince me. Is fine that the theoretical model include many approximations to be close to the experimental data, however these approximations must always be reported. If the final version still does show marginal similarity between the experimental and theoretical data, this is a point that has to be question. To me they cannot claim their conclusions with this disagreement between theory and experiment.

We thank the referee for the question but kindly disagree with the point. We think that the agreement between the calculations and the experiment capture the main aspects and validate our conclusions.

- (1) We agree with the referee that the theoretical angular dependence shows some deviation with respect to the experiment, however it captures the trend that we experimentally observe. For example, we show in Fig. 2 an overlay between the theoretical (solid lines) and experimental (error-bars) angular dependence for pi polarized incident light. We argue that our calculation predicts the essential behaviour such as: (a) the intensity of the single- and double-magnons is maximized at grazing incidence which is clearly not the case of the triple magnon. (b) the intensity of the single-magnon is minimized as the rotation angle is increased. (c) the intensity of the double-magnon is minimum at $\sim 80^\circ$. We acknowledge that it is difficult to predict where the maximum intensity of the triple-magnon is from our theory, however this mismatch does not render it useless as it successfully captures many other aspects as explained above.

Figure 2: Angular dependence of the magnons measured (error-bars) and computed (solid line) for π polarization. (A) Single-magnon, (B) double-magnon and (C) triple-magnon excitations measured at the incidence energy E1.

- (2) The deviation of the calculated angular dependence from the experiment can be due to several factors:
- (i) On the experimental side, as we performed our measurements on a bulk single crystal, we are prone to self-absorption and saturation effects. As the RIXS cross-section is a combination of an absorption (photon-in) process and an emission (photon-out) process, two geometrical effects have to be taken into account here: the probing depth is dependent on the x-ray absorption spectroscopy (XAS) cross-section (saturation) and the emitted photons can be re-absorbed (self-absorption). Consequently, the RIXS intensity is distorted in a bulk crystal according to the photon energy and the experimental geometry¹. As we rotate the sample, the probing depth is changed, and the photons emitted at different energies have different escape lengths, hence distorting the angular dependence. It is difficult to correct for these geometrical energy-dependent effects because it is affected by the background absorption (which is the off-resonant contribution from other elements in the sample and in the path of the beam).
 - (ii) On the theoretical side, one likely reason for the deviation could be the fact that the Fe sites in hematite have a small trigonal distortion. As a first approximation, the trigonal distortion would not change the ground state of Fe³⁺ because the singlet ⁶A₁ ground state does not split. However, the trigonal distortion would influence the intermediate states, and hence could modify the intensity and consequently the angular dependence.
 - (iii) Finally, we would like to point out that ligand field-multiplet theory reduces a full solid to a local cluster. This means that any Fe-Fe interactions or inter-cluster hopping are not considered. We expect that these effects are minimal in hematite, however, it is an approximation that can affect the angular dependence.
- (3) Comparing our results with published *2p3d* RIXS angular dependence such as that of the expert RIXS groups of M. Dean and M. Mitrano² conveys the level of accuracy expected by state-of-the-art theoretical modelling of a complex experiment such as RIXS in transition metal oxides. We show in Fig. 3 their experimental versus calculated angular dependence of Ni in La₄Ni₃O₈ using ligand field multiplet calculations. It is clear by comparing panels (a) and (b) that reproducing the full angular dependence (with the exact intensities) is usually not possible. This is because the computational model (and the experiment) has limitations as we discussed in the previous points. This however does not render the theoretical model useless as predicting a general trend is still very helpful as it provides insight about the main interactions at play and guides our understanding of experimental data.

¹Wang, et al., "Saturation and self-absorption effects in the angle-dependent *2p3d* RIXS spectra of Co³⁺," *J. Synchrotron Rad.*, vol. 27, p. 1–9, 2020.

² Shen et al., "Role of Oxygen States in the Low Valence Nickelate La₄Ni₃O₈," *Phys. Rev. X*, vol. 12, p. 011055, 2022.

Figure 3: Angle dependence of $\text{La}_4\text{Ni}_3\text{O}_8$ RIXS intensity at the preedge with σ polarization. (a) RIXS map data after absorption correction. (b) Simulation of a Ni_2O_7 cluster. (c) Orbital energy-level diagram of the multiplet ligand field. This figure is adapted from Ref².

- (4) Furthermore, the fact that the origin of the higher-rank magnons is the crystal field which acts as a momentum reservoir can be confirmed through our Feynman-diagrams. We show in Fig 4. the Feynman-diagrams for a $2p3d$ RIXS process in the absence of crystal field. It is evident in this case that only single- and double-magnons can be excited. Comparing Fig. 4 in this reply to Fig. 4 of the manuscript pinpoints that the origin of the higher-rank magnons is the crystal field and confirms theoretically the main conclusion of our paper.

Figure 4: Schematic representation of the mechanism of single- and double-magnons by $2p3d$ RIXS in the absence of crystal-field. The initial state vector is shown in the upper left corner comprising of the $2p$ (gray) and $3d$ (yellow) orbitals participating in the RIXS process. The spin of the electrons is depicted by the colored arrows (red = down, blue = up). We follow the fate of a $2p \rightarrow 3d$ excitation created by the absorption of a left polarized photon (A) through a cascade of $2p$ spin-orbit coupling and $2p$ - $3d$ exchange interaction through the steps from (B) to (E). It can be concluded that only double-magnons can be excited in the absence of crystal-field.

(5) In Fig. 5 we show a 3x3 matrix of the RIXS spectra calculated for the three possible photon polarization combinations (Left_{in}, Right_{in}, Z_{in} – Left_{out}, Right_{out}, Z_{out}). Our quantitative calculations confirm the selection rules we deduced from the Feynman-diagrams. These are: Upon the absorption of a left_{in} polarized photon, the system can (i)- decay elastically emitting a left_{out} polarized photon, (ii)- can excite a single-magnon emitting a Z_{out} polarized photon (iii)- can excite a double-magnon emitting a Right_{out} polarized photon. A similar figure for the crystal field case is presented in the supplementary and confirms the conclusion of Feynman-diagrams presented in the manuscript. The agreement between our Feynman diagrams predictions and the quantitative calculations confirm that our model captures the mechanism of multi-magnons excitations correctly.

Figure 52: $Fe^{3+} 2p3d$ calculations performed without taking into consideration crystal field. The 3x3 matrix shows the RIXS cross-section for the possible polarization combinations (Left, Right and Z polarized photons).

We stress that the magnitudes of the single-, double- and triple-magnons are correctly predicted by our model which implies

We have extended the discussion about the approximations of the calculations and the possible reasons for the observed deviation (lines 280-298). We also added Figures 2 and 3 to the supplementary as Fig. S5. The corresponding text referring to the supplementary figure are lines 229-231 in the manuscript.

Overall, I do see a nice report of the experimental observance of the 3rd and 4th magnons. However I do see disagreement between the theory and the experiments, which make their bold conclusions to be questionable. Also, I find their decision to remove the experimental data that brings up these questions to be very inadequate.

We hope that our answers above have clarified the discrepancy between the calculations and the experiment and that the referee finds our manuscript now suitable for publication.

Reviewer #3 (Remarks to the Author):

Dear Editor,

the authors addressed my previous questions. But I still have difficulties following the authors reasoning on the difference between local, single-site excitations of electrons and nonlocal, collective excitations such as magnons.

We acknowledge the point of the referee as RIXS can probe both local single-site and nonlocal collective excitations such as magnons. As hematite is an antiferromagnetic material, the magnetic excitations measured in an extended hematite crystal are naturally nonlocal collective excitations in the form of magnons. Fig. 6(a) shows inelastic neutron scattering measurements in hematite³. The \mathbf{q} point where we performed our measurements is marked with the red arrow. The magnon energy that we find in our RIXS measurement matches with the energy reported by neutron scattering which confirms our assignment of the first low energy peak as a single-magnon. We note that the experimental energy resolution of a RIXS measurement does not allow for resolving the optical and acoustic branches.

What we think causes confusion is the fact that a magnon is commonly interpreted to originate from a local single-site spin-flip process. This description is widely used because the intermediate state in a $2p3d$ RIXS experiment is strongly localized due to the $2p$ core-hole and hence such a local picture is very useful to describe many aspects of the RIXS process. However, the final state excited by RIXS is not necessarily localized and can be a collective excitation such as magnons in a magnetic material. The way to reconcile both the local single-site and collective aspects of RIXS is to realize that the incident photon can be scattered at any equivalent site, leading to a final state that is a superposition of spin-flips at equivalent sites. Such a final state carries a nonlocal magnetic excitation and represents the magnon density of states as also confirmed by detailed comparison to inelastic neutron scattering data. The fact that RIXS measures magnons in magnetic materials has been studied in great details where we show in Fig. 6(b) a comparison between the magnon dispersion measured in La_2CuO_4 by $2p3d$ RIXS (red dots) and inelastic neutron scattering (blue dashed line) confirming the capability of $2p3d$ RIXS to measure magnons⁴.

Figure 3: (a) Experimental magnon dispersion observed using inelastic neutron scattering (squares) in hematite overlaid with calculated dispersion relation (solid) adapted from Ref³. (b) Comparison between the dispersion of magnons in La_2CuO_4 measured by $2p3d$ RIXS (red circles) and inelastic neutron scattering (blue dashed lines) adapted from Ref⁴.

³ Samuelsen et. al., "Inelastic Neutron Scattering Investigation of Spin Waves and Magnetic Interactions in alpha-hematite," *Phys. Stat. Sol.*, vol. 42, p. 241, 1970.

⁴ H. Robarts, et al., "Dynamical Spin Susceptibility in La_2CuO_4 Studied by Resonant Inelastic X-ray Scattering," *Phys. Rev. B*, vol. 103, p. 224427, 2021.

Having identified the first magnetic excitation as a single-magnon, we now address the collective nature of the multi-magnons. The multi-magnons nearly do not disperse in three dimensional magnetic systems as predicted by the calculations of Haverkort⁵. Fig. 7 shows the calculated RIXS cross-section in NiO where one can observe single-magnons (Fig. 7(a) which shows visible dispersion as expected) and double-magnons (Fig. 7(b)). Even though the multi-magnons form a continuous band, the intensity is maximized at double the energy of a single-magnon in a form of a flat band. The fact that the multi-magnons nearly do not disperse however does not imply that they are local excitations just like the fact that the optical single-magnon mode shown in Fig. 6(a) nearly not dispersing does not imply that it is a local excitation. Both magnons, the optical single-magnon and multi-magnons, are collective excitations predicted by spin-wave theory considering many electrons.

Figure 4: 2p3d RIXS spectral function calculated using linear spin-wave theory for a Heisenberg model for (a) single-magnon and (b) double-magnon dispersion for Ni²⁺. The energy is given in units of zSJ (number of neighbors, spin, and exchange constant). This figure is adapted from Ref⁵.

We have clarified why we expect that the magnetic excitations that we observed are collective magnons as discussed. We have also clarified the terminology of single-site spin-flip excitation and a collective magnon excitation as discussed above. This can be found in lines 56 – 70 and 133 - 138 in the manuscript.

Maybe the reason of my confusion is because we are from a different community with different usage of words. Nevertheless, I think that the manuscript should be understandable for a larger community if published in Nature Communications. Hence, I have a question where I am believe that a good answer could create a bridge to a larger readership: The authors raise a fundamental question (Line 83-85): “Is it possible to change the spin angular momentum of a system with an amount greater than the change in the x-ray photon angular momentum of the RIXS experiment?”

In my research field, we also have multi-magnon scattering processes, where the multiple magnons scatter and annihilate and create new magnons. One famous example is three-magnon-scattering, where a the ferromagnetic resonance can decay into two new magnons with half the energy and opposite linear momentum. The spin momentum, however, is obviously not conserved (1 magnon in, two magnons out). It is generally accepted, that the break in rotational symmetry due to the non-isotropic magnetic-dipole-interaction is causing the breakdown of angular momentum conservation.

⁵ Haverkort, *Phys. Rev.Lett.*, vol. 105, p. 167404, 2010.

We thank the referee for this comment. As FMR is a popular technique to study magnetic excitations in materials, we agree that pointing out the main similarities and the differences between FMR and RIXS is an effective way to bridge a wide range of the community. While both techniques can probe magnons, below we highlight the key differences in between:

Differences between RIXS and FMR:

- The light-matter interaction in both techniques. While in FMR, one considers the coupling between the magnetic field of a microwave light with a magnetic material, in RIXS one considers the coupling between the electric field of an x-ray photon with a magnetic material.
- FMR measurements are performed in microwave resonant cavities where there can be a strong coupling between the cavity resonance modes and the magnons. This leads to a situation where the Fermi golden rule fails and results in the observation of hybridized excitations such as the magnon-polaritons. This is not the case for our RIXS experiments where the photon densities produced by synchrotrons are low (and are only partially coherent) and perturbation theory can be well applied.
- A microwave photon has nearly no linear momentum while in the x-ray regime photons have a significant linearmomentum.
- FMR can measure magnons only at $\mathbf{q}=0$ (gamma point, or at $-k$ and $+k$ pairs in the case of three magnon scattering) while the momentum transferred to the system from the photons is determined by $\mathbf{q}=\mathbf{k}_{out} - \mathbf{k}_{in}$ as shown in Fig 8. Magnon can be excited at $\mathbf{q}=\mathbf{k}_{out} - \mathbf{k}_{in}$ using RIXS.

Figure 5: (a) Scattering geometry in a RIXS experiment where an incident photon with a wavevector k_{in} is resonantly absorbed by the material and an outgoing photon with a wavevector k_{out} is resonantly emitted. The difference: $\mathbf{q} = \mathbf{k}_{out} - \mathbf{k}_{in}$ is the transferred momentum which in the x-ray regime is non-negligible. (b) A sketch showing the two steps of RIXS: resonant absorption of an incident photon (electric dipole transition) followed by an emission of an outgoing photon (electric dipole transition). The energy and momentum difference between the incident and outgoing photon can excite a magnon.

- For both spectroscopies, the selection rules considering the spin (and orbital) angular momenta are strictly applied only when the rotational symmetry is preserved. When the rotational symmetry is broken, the selection rules can be broken. In a $2p3d$ RIXS experiment where the light-matter interaction is dominantly an electric dipole, the relevant symmetry breaking arises from the electrostatic non-spherical field induced by the neighboring atoms (i.e., crystal field).

We conclude that what is remarkable about our experiment, is that the cross-section of the spin non-conserving excitations (three, four, and five multi-magnons) are enhanced due to the resonant effect.

We have pointed briefly some of the differences between FMR and RIXS in the manuscript in lines 67 – 70.

Also, keep in mind that a magnon is not only the excitation of the spin-system. Even magnons with high wave-vectors are still polaritons, meaning they are a hybridized excitation of a nonlocal, collective spin flip AND an electromagnetic wave coupled to the magnetic excitation.

The referee points out one of the differences between FMR and RIXS. While FMR measurements are performed in microwave cavities where there is an opportunity to have a strong coupling between the photons and the magnons thanks to the coherence provided by the cavity, this is not the case in a RIXS experiment because the photon densities produced by synchrotrons are low (and are only partially coherent). Our experiment can be well described in the weak coupling regime where this hybridization between the light and matter doesn't occur. The light-matter interaction can be treated by perturbation theory for a RIXS experiment measured in a synchrotron.

Here is my fundamental question: If multi-magnons are excited in the authors experiment and if this is indeed a non-local, collective excitation of several electrons, then there should be a magnetic-dipole field associated with this excitation, which also breaks the angular momentum conservation in scattering processes with X-rays.

We do not think that multi-magnons will give rise to magnetic dipole fields that break the momentum conservation in the scattering process with the x-rays. This is because at this energy range, the dominant light-matter interaction is through the electric dipole and not the magnetic-dipole.

Reviewers' Comments:

Reviewer #1:

Remarks to the Author:

Dear Editor,

The authors made quite some efforts addressing my questions and the questions by Reviewer #1. However, I feel the same way as Reviewer #1 that this article might fit better in a more specialized journal.

I tried to build a bridge in understanding the presented data by mentioning some analogies with other methods. But the authors separate the article even further. Ferromagnetic resonance doesn't couple to other magnons only because it's a cavity. Most FMR measurements are actually performed not in the regime of strong coupling. This would go in the direction of quantum-magnonics where a large cooperatively between a cavity mode and the magnon mode is observed. And even if magnons are excited by photons in the visible range, they are still polaritons. If the authors consider a single-site spin flip, then they observe Stoner-excitations, and not magnons. Nowadays FMR is typically done without a microwave cavity simply with a broadband stripe-line and can even be performed at non-zero wave vectors, which is called "propagating spin wave spectroscopy". And there are also laser-spectroscopy methods such as Brillouin light scattering and time-resolved magneto-optical Kerr-effect microscopy. Both methods allow for measuring the dispersion of magnons.

Reviewer #2:

Remarks to the Author:

The authors have thoroughly addressed all my concerns. I'm happy to recommend the manuscript to be published in the current form.

Reply to Reviewer comments on:

Novel magnetic excitations beyond the single- and double-magnons

Hebatalla Elnaggar,^{1,2*} Abhishek Nag,³ Maurits W. Haverkort,⁴ Mirian Garcia-Fernandez,³
Andrew Walters,³ Ru-Pan Wang,^{1,5} Ke-Jin Zhou,^{3*} Frank de Groot^{1*}

¹Debye Institute for Nanomaterials Science, Utrecht University, 3584 CA Utrecht, The Netherlands.

²Institute of Mineralogy, Physics of Materials and Cosmochemistry, CNRS Sorbonne University, 4
Place

Jussieu, 75005 Paris, France.

³Diamond Light Source, Harwell Campus, Didcot OX11 0DE, United Kingdom.

⁴Heidelberg University, Philosophenweg 19, 69120 Heidelberg, Germany.

⁵Department of Physics, University of Hamburg, Luruper Chaussee 149, G610, 22761
Hamburg, Germany.

Reviewer comments are in black

Authors reply is in blue

REVIEWERS' COMMENTS

Reviewer #1 (Remarks to the Author):

Dear Editor,

The authors made quite some efforts addressing my questions and the questions by Reviewer #1. However, I feel the same way as Reviewer #1 that this article might fit better in a more specialized journal.

I tried to build a bridge in understanding the presented data by mentioning some analogies with other methods. But the authors separate the article even further. Ferromagnetic resonance doesn't couple to other magnons only because it's a cavity. Most FMR measurements are actually performed not in the regime of strong coupling. This would go in the direction of quantum-magnonics where a large cooperatively between a cavity mode and the magnon mode is observed. And even if magnons are excited by photons in the visible range, they are still polaritons. If the authors consider a single-site spin flip, then they observe Stoner-excitations, and not magnons. Nowadays FMR is typically done without a microwave cavity simply with a broadband stripe-line and can even be performed at non-zero wave vectors, which is called "propagating spin wave spectroscopy". And there are also laser-spectroscopy methods such as Brillouin light scattering and time-resolved magneto-optical Kerr-effect microscopy. Both methods allow for measuring the dispersion of magnons.

We removed the reference to FMR only probing \$q=0\$ magnons in line 69.

Reviewer #2 (Remarks to the Author):

The authors have thoroughly addressed all my concerns. I'm happy to recommend the manuscript to be published in the current form.

We thank the referee for their input and comments that improved our paper and are happy that our manuscript is now suitable for publication.